

# DNA methylation profiling in recurrent miscarriage

Li Pi[1,*], Zhaofeng Zhang[1,*], Yan Gu[2], Xinyue Wang[1], Jianmei Wang[2], Jianhua Xu[1], Junwei Liu[1], Xuan Zhang[1] and Jing Du[1]

[1] NHC Key Lab. of Reproduction Regulation (Shanghai Institute of Planned Parenthood Research), Medical School, Fudan University, Shanghai, China
[2] The Second Hospital of Tianjin Medical University, Tianjin, China
[*] These authors contributed equally to this work.

## ABSTRACT

Recurrent miscarriage (RM) is a complex clinical problem. However, specific diagnostic biomarkers and candidate regulatory targets have not yet been identified. To explore RM-related biological markers and processes, we performed a genome-wide DNA methylation analysis using the Illumina Infinium HumanMethylation450 array platform. Methylation variable positions and differentially methylated regions (DMRs) were selected using the Limma package in R language. Thereafter, gene ontology (GO) enrichment analysis and pathway enrichment analysis were performed on these DMRs. A total of 1,799 DMRs were filtered out between patients with RM and healthy pregnant women. The GO terms were mainly related to system development, plasma membrane part, and sequence-specific DNA binding, while the enriched pathways included cell adhesion molecules, type I diabetes mellitus, and ECM–receptor interactions. In addition, genes, including *ABR*, *ALCAM*, *HLA-E*, *HLA-G*, and *ISG15*, were obtained. These genes may be potential candidates for diagnostic biomarkers and possible regulatory targets in RM. We then detected the mRNA expression levels of the candidate genes. The mRNA expression levels of the candidate genes in the RM group were significantly higher than those in the control group. However, additional research is still required to confirm their potential roles in the occurrence of RM.

## INTRODUCTION

Recurrent miscarriage (RM), defined as two or more consecutive clinically recognized spontaneous pregnancy losses before 20 weeks of gestation (*Rai & Regan, 2006*), affects 1–5% of women within the reproductive age (*Kim et al., 2004*; *Pildner & Ktm, 2009*). Multiple causes have been found to contribute to the pathogenesis of RM, including chromosomal abnormalities, cervical incompetence, uterine anomalies, autoimmune diseases, endocrinological abnormalities, antiphospholipid antibodies, thrombophilic disorders, low progesterone levels, and microbial infections (*Hou et al., 2016*; *Rai & Regan, 2006*). Approximately 40–50% of cases remain unexplained (*Li et al., 2002*); however, the molecular mechanisms have not been fully identified (*Griebel et al., 2005*). Determining

Corresponding authors
Xuan Zhang, xuanzhang@sippr.org.cn
Jing Du, dujing42@126.com

potential diagnostic biomarkers and possible regulatory targets of RM may help promote research progress. Therefore, new methods are needed.

Recently, there has been an increasing interest in the role of epigenetic mechanisms in human diseases. One promising approach is DNA methylation profiling. DNA methylation, a well-characterized epigenetic modification, is critical for development and differentiation (*Li, Bestor & Jaenisch, 1992*; *Ziller et al., 2013*). In addition, it has been proposed that DNA methylation may be an important factor in the regulation of gene expression, X chromosome inactivation, genomic imprinting, chromatin modification, endogenous retrovirus silencing, and developmental origins of common human diseases (*Bestor, 2000*; *Bird & Wolffe, 1999*; *Reik & Walter, 2001*; *Takai & Jones, 2002*). By using DNA methylation profiling, we can obtain adequate information on aberrant DNA methylation events (*Yagi et al., 2008*).

In addition, a previous study has established that DNA methylation occurs during early embryonic development (*Li, 2002*). Aberrant DNA methylation, arising during embryonic development, has been identified as a potential cause of pregnancy loss (*Hanna, McFadden & Robinson, 2013*). Accordingly, we mainly focused on the genes involved in embryonic development to identify the biomarkers of RM.

In the present study, we used the Illumina Infinium HumanMethylation450 array platform to conduct a genome-wide screening of DNA methylation in decidua samples from the products of conception of women with RM and to identify novel methylation variable positions (MVPs) and differentially methylated regions (DMRs). Furthermore, to gain insight into the molecular regulatory mechanisms of RM, gene ontology (GO) and pathway enrichment analyses were used to explore the potential diagnostic biomarkers and possible regulatory targets. The mRNA expression levels of candidate genes were detected using real-time PCR.

The contribution of errors to miscarriage and whether the DNA methylation process plays a pivotal role in fetal programming have not been well explored. In this study, we aimed to evaluate the association of aberrant DNA methylation between patients with RM and healthy pregnant women and obtain data for further understanding human pregnancy loss by predicting potential RM-related biological processes and pathways, as well as candidate genes.

## MATERIAL AND METHODS

### Samples

We enrolled 15 healthy pregnant women and 15 patients with RM from the outpatient department of Gynecology and Obstetrics, The Second Hospital of Tianjin Medical University, China. All participants were recruited in accordance with the same inclusion and exclusion criteria. The participants were considered to have RM if they had at least two consecutive miscarriages. The participants were considered as controls if they had at least one live birth and no history of miscarriage, still birth, preterm labor, or pre-eclampsia and if their pregnancy was terminated for non-medical reasons or they underwent legal abortions. The exclusion criteria included endocrine diseases, infections, chromosomal

**Table 1  Clinical characteristics of the recruited recurrent miscarriage (RM) patients ($n = 15$) and normal pregnant (NP) women ($n = 15$).**

|  | RM (mean ± SD) | NP (mean ± SD) |
|---|---|---|
| Age(years) | 30.13 ± 4.42 | 29.53 ± 6.65 |
| Gestational weeks | 8.27 ± 1.87 | 7.33 ± 0.82 |
| Childbearing history | 0.27 ± 0.46 | 0.53 ± 0.52 |
| Spontaneous abortion history | 2.53 ± 0.64 |  |

abnormalities, immunological diseases, and anatomical abnormalities of the genital tract. Gestational weeks ranged from 6 to 12 weeks, and the maternal age ranged from 22 to 42 years (Table 1). Decidual tissue was collected anonymously via curettage in The Second Hospital of Tianjin Medical University from June 2013 to August 2013. All collected tissues were fragmented and stored immediately at −80 °C in RNAlater solution (Invitrogen, Carlsbad, CA) until RNA and DNA extraction. Written informed consent was obtained from all patients who provided tissue samples, and consent to publish research data derived from these collected samples was also obtained. The collection of the samples for this study was approved by the Shanghai Institute of Planned Parenthood Research Clinical Ethics Review Board (No.PJ2015).

## DNA extraction

According to the manufacturer's protocol, genomic DNA was extracted from the decidual tissues using the DNeasy Blood and Tissue Kit (TransGen, China) after the samples were digested via proteinase K and treated with RNase.

## Infinium HumanMethylation450 BeadChip processing

DNA from each sample was treated with sodium bisulfate and processed for analysis on the Illumina Infinium HumanMethylation450 array platform at Genergy (Shanghai, China). Before proceeding to the statistical analysis, all data were processed via Beta Mixture Quantile dilation. Linear models were developed to calculate P-values using the Limma package of the R software (*Smyth, 2005*). After Benjamini and Hochberg correction, we screened significant MVPs. Meanwhile, using the Probe Lasso method of the ChAMP package, we estimated the DMRs. Volcano plot and heat map software programs were used to analyze and visualize the data.

## GO enrichment analysis

GO analysis was utilized to explain the primary DMR function based on the GO database, which is the crucial functional classification database of the NCBI (*Ashburner et al., 2000*; *Gene Ontology, 2006*). Fisher's exact test was used to calculate the significance level (*P*-value) of each GO term to screen out the significant GO terms of DMR enrichment. *P*-values <0.05 were considered to be statistically significant.

## Pathway enrichment analysis

Pathway analysis was performed to determine the significant pathway terms of the DMRs according to the Kyoto Encyclopedia of Genes and Genomes (KEGG)
(*Kanehisa & Goto, 2000*). We used Fisher's exact test to identify significant pathway terms, and *P*-values <0.05 were also considered statistically significant (*Draghici et al., 2007*; *Kanehisa et al., 2004*).

## Quantitative real-time PCR

The total RNA of the decidual tissues was extracted using the EasyPure RNA Kit (TransGen Biotech, Beijing, China). Reverse transcription reaction was performed with 500 ng of total RNA using the PrimeScript RT Master Mix Perfect Real Time Kit (TaKaRa, Japan). cDNA (2 µL) was added to the 18 µL reaction mixture containing 6 µL of ddH$_2$O, 10 µL of SYBR Premix Ex Taq II, 0.4 µL of ROX Reference Dye or Dye II (TaKaRa), and 0.8 µL of each primer. The PCR conditions were as follows: 95 °C for 1 min, followed by 40 cycles at 95 °C for 15 s, 58 °C for 20 s, and 72 °C for 20 s, and a final extension at 72 °C for 5 min. All samples were assayed in triplicate. Relative gene expression levels were calculated using the $2^{-\Delta\Delta}$ method.

# RESULTS

## Volcano plot and heat map of the MVPs

We used the R software to screen the MVPs. Using the algorithms provided in the Limma package after Benjamini and Hochberg correction, we collected data and produced a volcano plot and heat map.

The volcano plot, generated using the volcano plot function, showed the results of the MVPs between the controls and patients with RM. The dots in the upper left section denoted the hypo-methylated loci, and those in the upper right section denoted the hyper-methylated loci (Fig. 1A). The heat map, generated using the heatmap function, showed the hierarchical clustering of the MVPs between the controls and patients with RM (Fig. 1B).

The MVP analysis indicated that C2orf54, CECR2, TMEM161B, ABR, EMCN, TBCD, GP5, PRICKLE2-AS3, NBPF22P, and TBX3 showed differential methylation levels with respect to the cases and controls (Table 2). Among these genes, we screened CECR2 (*P* = 1.7299E−06), ABR (*P* = 3.55356E−06), and TBX3 (*P* = 6.52408E−06), which are closely related to embryonic development, as candidate genes of RM (*Banting et al., 2005*; *Chen et al., 2010*; *Esmailpour & Huang, 2012*; *Ohgushi et al., 2017*).

## GO enrichment analysis and GO-Tree of the DMRs

To conduct a functional enrichment analysis to identify RM-related biological processes, cellular components, and molecular functions, GO enrichment analysis of 1,799 DMRs was conducted. The most enriched biological processes as determined in the GO term analysis were associated with system development (GO:0048731, *P* = 1.86E−30), multicellular organismal development (GO:0007275, *P* = 1.05E−29), and anatomical structure development (GO:0048856, *P* = 4.62E−28). Within the cellular component category, the most enriched GO terms were significantly associated with the plasma membrane part (GO:0044459, *P* = 9.42E−11), plasma membrane (GO:0005886, *P* = 1.86E−09), and cell periphery (GO:0071944, *P* = 4.57E−09). In the molecular function category, the GO terms

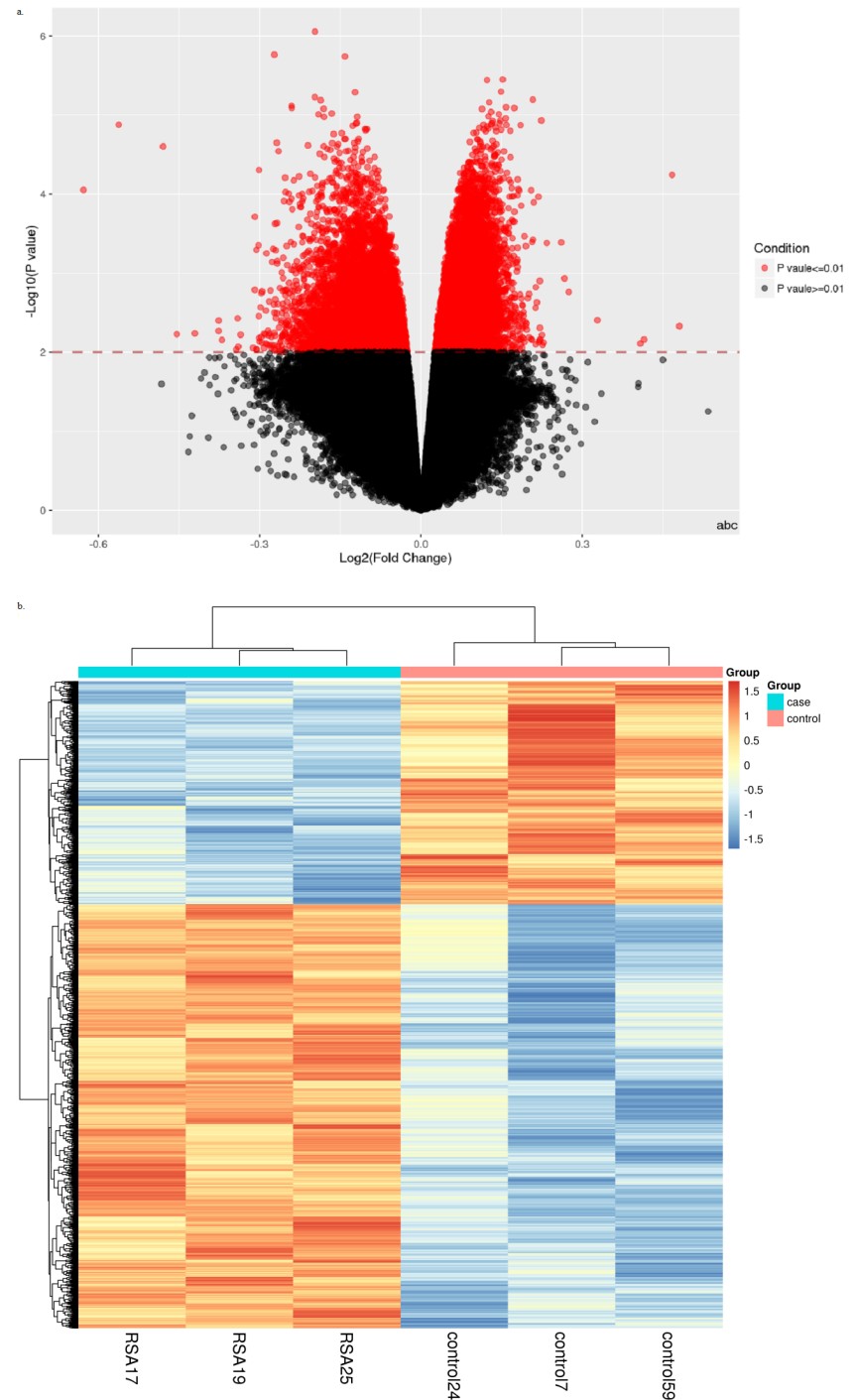

**Figure 1  Viewsof methylation variable positions (MVPs) result.** (A) Volcano plot of MVPs between control and RM. Each dot represents an individual MVPs. Dots that showed *P* values <= 0.01 after Benjamini & Hochberg correction are colored red while *P* values >= 0.01 after Benjamini & Hochberg correction are colored black. The *x*-axis represents log2 (Fold Change). The *y*-axis shows the −log10 (*p* value). (B) Heat map of MVPs between control and RM. Each row represents a locus and each column represents a sample. Red indicates that the methylation level is up-regulated and blue indicates down-regulated methylation level.

**Table 2  The methylation variable positions (MVPs).**

| gene | probeID | logFC | t | P.Value |
|------|---------|-------|---|---------|
| C2orf54 | cg06546066 | 0.209665031 | −23.50980689 | 8.83141E-07 |
| CECR2 | cg27415324 | −0.260083489 | −20.80755786 | 1.7299E-06 |
| TMEM161B | cg18295770 | 0.101275636 | −20.62487447 | 1.81586E-06 |
| ABR | cg09639964 | −0.590951212 | 18.25084313 | 3.55356E-06 |
| EMCN | cg13149566 | 0.419028173 | 18.19585048 | 3.61283E-06 |
| TBCD | cg06218079 | −0.051208796 | 17.10132934 | 5.07438E-06 |
| GP5 | cg10658438 | 0.216737391 | −17.05295333 | 5.15363E-06 |
| PRICKLE2-AS3 | cg05063999 | −0.013472989 | −16.61485758 | 5.94177E-06 |
| NBPF22P | cg12475092 | 0.007591767 | 16.38637216 | 6.40879E-06 |
| TBX3 | cg17070988 | 0.562046907 | −16.33297277 | 6.52408E-06 |

enriched for the DMRs in RM included sequence-specific DNA binding (GO:0043565, $P = 1.61\text{E}{-}24$), sequence-specific DNA binding transcription factor activity (GO:0003700, $P = 7.89\text{E}{-}21$), and nucleic acid binding transcription factor activity (GO:0001071, $P = 9.29\text{E}{-}21$) (Fig. 2A, Table 3).

The GO enrichment analysis for the DMRs indicated that the genes were mainly enriched in the GO terms, including *HES4, PRKCZ, SKI, PRDM16, ALPL, RUNX3, TAL1, TTLL7, BARHL2, OLFM3, NFYC, TBX15, LMX1A, TNFRSF4, AJAP1, SYTL1, OPRD1, PTPRU, PRRX1, ELF3, PROX1, HLA-G,* and *HLA-E* (Table 4). Among these, we screened *HLA-G* ($P = 1.86\text{E}{-}30$), *HLA-E* ($P = 9.42\text{E}{-}11$), and *PRDM16* ($P = 1.86\text{E}{-}30$), which are closely related to embryonic development, as candidate genes of RM (*Gelmini et al., 2016*; *Horn et al., 2011*; *Verloes et al., 2017*).

To determine the intrinsic link among gene functions, hierarchical trees were constructed. The results showed that the genes were closely related to development, plasma, and DNA binding in RM (Figs. 2B, 2C and 2D).

## Pathway enrichment analysis of the DMRs

To generate further insight into the pathways, we performed KEGG pathway annotation to determine whether significant DMRs were enriched for any pathway. From the results, the top five frequently enriched pathways were as follows: cell adhesion molecules (CAMs) (PATHWAY:hsa04514, $P = 3.00\text{E}{-}08$), type I diabetes mellitus (PATHWAY:hsa04940, $P = 1.27\text{E}{-}06$), ECM-receptor interaction (PATHWAY:hsa04512, $P = 4.24\text{E}{-}06$), allograft rejection (PATHWAY:hsa05330, 1.02E-05), and antigen processing and presentation (PATHWAY:hsa04612, $p = 2.05\text{E}{-}05$) (Fig. 3, Table 5).

The KEGG pathway enrichment analysis for the DMRs indicated the mainly enriched genes, which included *CADM3, F11R, NRXN1, ITGA4, PDCD1, ALCAM, NLGN1, CLDN16, HLA-G, HLA-E, HLA-B, TNF, HLA-DQB1, COL11A1, SV2A, LAMB3, ITGA4,* and *COL6A3* (Table 6). Among these, we screened *ALCAM* ($P = 3.00\text{E}{-}08$), *HLA-G* ($P = 3.00\text{E}{-}08$), and *HLA-E* ($P = 3.00\text{E}{-}08$), which are closely related to embryonic development, as candidate genes of RM (*Cizelsky & Tata, 2014*; *Gelmini et al., 2016*; *Verloes et al., 2017*).

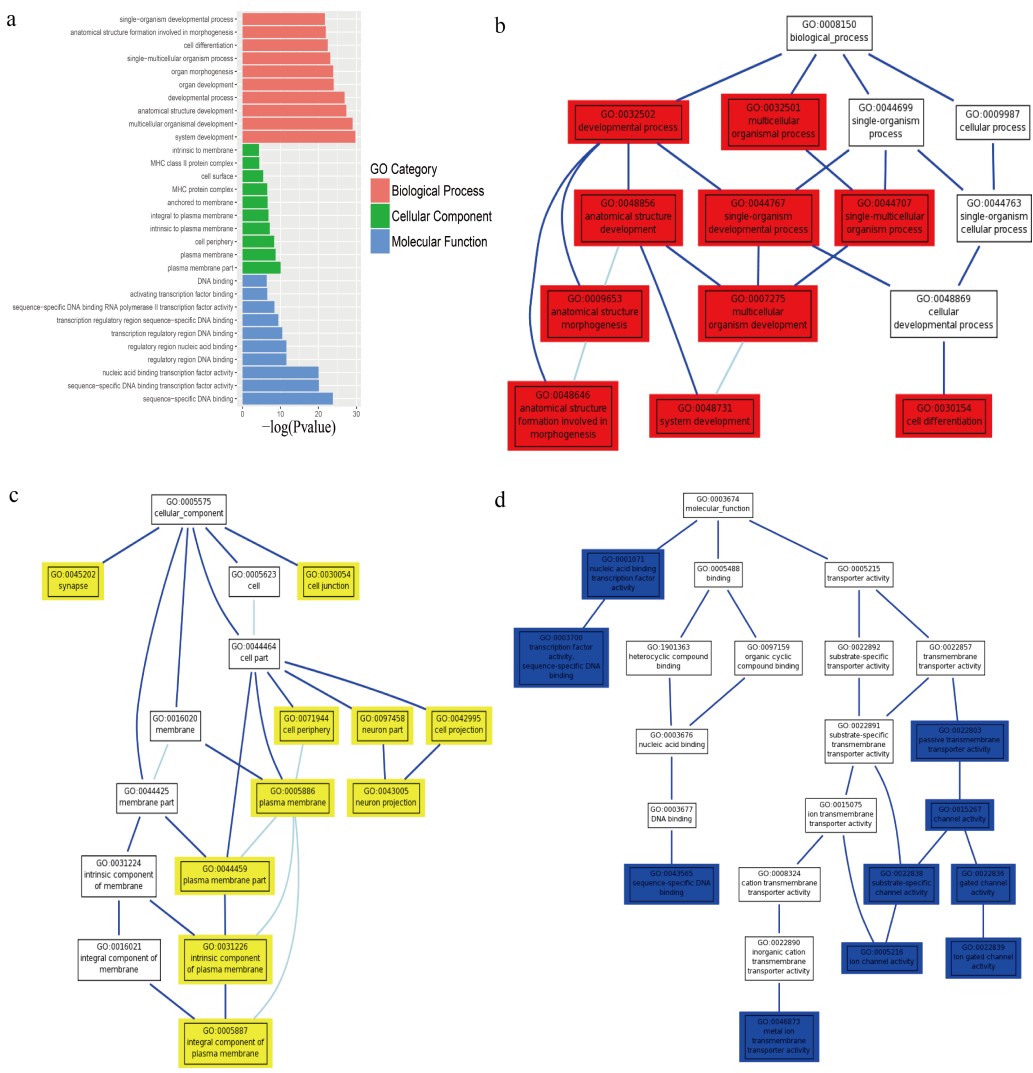

**Figure 2  GO enrichment analysis and GO-Tree of differentially methylated regions (DMRs).** (A) GO enrichment analysis. The horizontal axis was the value of $-\log(P\text{-value})$. The vertical axis was GO terms colored in three levels. The most significantly enriched terms for each level are shown. Red represented biological process (BP), green represented cellular component (CC), and blue represented molecular function (MF). (B–D) GO-tree. Hierarchical trees of three categories represented biological process (BP), cellular component (CC) and molecular function (MF) respectively. Colored squares represented the most enriched GO terms.

## mRNA relative expression level of the candidate genes in 24 samples

Based on the abovementioned processing and analysis data, we selected five genes (*ISG15*, *ABR*, *HLA-E*, *HLA-G*, and *ALCAM*) with significant differences in methylation expression and possible relationships to embryogenesis and development for quantitative real-time PCR verification. The analysis showed that the mRNA expression levels of *ISG15*, *ABR*, *HLA-E*, and *HLA-G* were higher in the RM group than in the control group ($P < 0.05$). There was no significant difference in the mRNA expression level of *ALCAM* between the two groups ($P > 0.05$) (Fig. 4).
**Table 3  Gene ontology (GO) Enrichment analysis (top 3 significantly enriched biology terms).**

| ID | Category | Term | *p*-value |
|---|---|---|---|
| biological process (BP) | GO:0048731 | system development | 1.86E-30 |
| | GO:0007275 | multicellular organismal development | 1.05E-29 |
| | GO:0048856 | anatomical structure development | 4.62E-28 |
| cellular component (CC) | GO:0044459 | plasma membrane part | 9.42E-11 |
| | GO:0005886 | plasma membrane | 1.86E-09 |
| | GO:0071944 | cell periphery | 4.57E-09 |
| molecular function (MF) | GO:0043565 | sequence-specific DNA binding | 1.61E-24 |
| | GO:0003700 | sequence-specific DNA binding transcription factor activity | 7.89E-21 |
| | GO:0001071 | nucleic acid binding transcription factor activity | 9.29E-21 |

**Table 4  GO-related genes.**

| GO_term | GO_category | symbols_in_list | P value |
|---|---|---|---|
| system development | biological_process | HES4,PRKCZ,SKI,PRDM16,ALPL,RUNX3,TAL1,TTLL7, BARHL2,OLFM3,COL11A1,TBX15,WARS2,HLA-G | 1.86E-30 |
| plasma membrane part | cellular_component | TNFRSF4,PRKCZ,AJAP1,SYTL1,OPRD1,PTPRU,1L12RB2, GJA5,GNG4,GPR137B,SNTG2,TPO,HLA-G,HLA-E | 9.42E-11 |
| sequence-specific DNA binding | molecular_function | PRDM16,RUNX3,NFYC,TAL1,BARHL2,TBX15,LMX1A, PRRX1,ELF3,PROX1,SIX3,OTX1,MEIS1,VAX2,EN1 | 1.61E-24 |

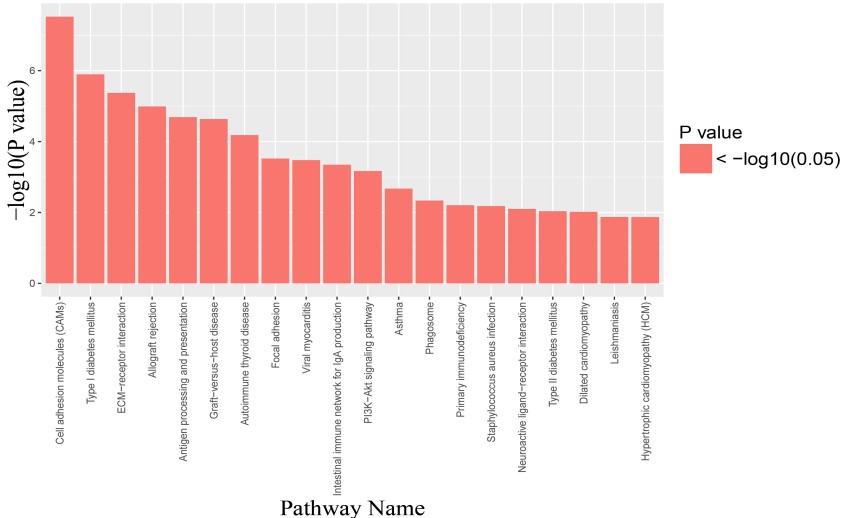

**Figure 3  Pathway Enrichment Analysis of differentially methylated regions (DMRs).** KEGG pathway enrichment analysis. The horizontal axis was the top 20 most significantly enriched KEGG pathway. The vertical axis was the value of −log(*P*-value).

**Table 5  KEGG enrichment analysis (top 10 significantly enriched pathway terms).**

| ID | Category | Term | *p*-value |
|---|---|---|---|
| EGG PATHWAY | PATHWAY:hsa04514 | Cell adhesion molecules (CAMs) | 3.00E-08 |
| | PATHWAY:hsa04940 | Type I diabetes mellitus | 1.27E-06 |
| | PATHWAY:hsa04512 | ECM-receptor interaction | 4.24E-06 |
| | PATHWAY:hsa05330 | Allograft rejection | 1.02E-05 |
| | PATHWAY:hsa04612 | Antigen processing and presentation | 2.05E-05 |
| | PATHWAY:hsa05332 | Graft-versus-host disease | 2.32E-05 |
| | PATHWAY:hsa05320 | Autoimmune thyroid disease | 6.56E-05 |
| | PATHWAY:hsa04510 | Focal adhesion | 3.02E-04 |
| | PATHWAY:hsa05416 | Viral myocarditis | 3.34E-04 |
| | PATHWAY:hsa04672 | Intestinal immune network for IgA production | 4.50E-04 |

**Table 6  KEGG-related genes.**

| Pathway_name | Pathway_class | Symbols_in_list | *P* value |
|---|---|---|---|
| Cell adhesion molecules (CAMs) | Signaling molecules and interaction | CADM3,F11R,NRXN1,ITGA4,PDCD1,ALCAM,NLGN1,CLDN16, HLA-G,HLA-E,HLA-B,HLA-DQB1,HLA-DOB,HLA-DMB | 3.00E-08 |
| Type I diabetes mellitus | Endocrine and metabolic diseases | HLA-G,HLA-E,HLA-B,TNF,HLA-DQB1,HLA-DOB,HLA-DMB,HLA-DMA,HLA-DOA,HLA-DPB1,PTPRN2,PRF1,INS | 1.27E-06 |
| Allograft rejection | Immune diseases | HLA-G,HLA-E,HLA-B,TNF,HLA-DQB1,HLA-DOB, HLA-DMB,HLA-DMA,HLA-DOA,HLA-DPB1,PRF1 | 1.02E-05 |
| Antigen processing and presentation | Immune system | NFYC,HLA-G,HLA-E,HLA-B,TNF,HLA-DQB1, HLA-DOB,TAP2,TAP1,HLA-DMB,HLA-DMA,HLA-DOA | 2.05E-05 |
| Focal adhesion | Cell communication | COL11A1,LAMB3,ITGA4,COL6A3,ITGA2,PDGFRB,FLT4,MYLK4, TNXB,COL11A2,THBS2,EGFR,ITGA8,DOCK1,CCND1,CCND2 | 3.02E-04 |

## DISCUSSION

Previous studies have attempted to determine the association between the DNA methylation status and mechanisms that lead to RM. Some studies have suggested that *MTHFR* is a candidate gene that plays an important role during pregnancy by regulating thrombotic events or methylation (*Mishra et al., 2019*). Other studies have shown that increasing FOXP3 promoter methylation levels may cause abnormal immune tolerance through downregulation of the expression of FOXP3 protein, which consequently leads to unexplained recurrent spontaneous abortion (*Hou et al., 2016*). A study in which combined analysis of DNA methylation and gene expression was performed showed CREB5 as a contributor in RM (*Yu et al., 2018*). Moreover, abnormal methylation of the decidua has been demonstrated to be associated with pregnancy failure in an animal model (*Brown et al., 2013*). In this study, we analyzed the DNA methylation profile of MVPs and DMRs in decidua samples obtained from 15 patients with RM and 15 controls via a genome-wide DNA methylation analysis using the microarray platform, Illumina Infinium HumanMethylation450 BeadChip. Furthermore, we were able to conduct GO enrichment analysis and pathway enrichment analysis of the DMRs. In the GO enrichment analysis,

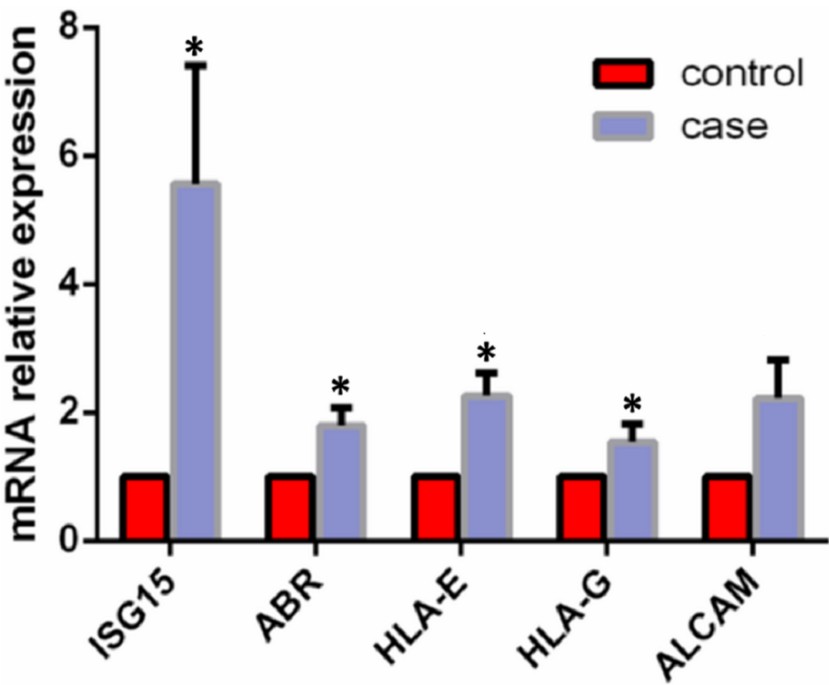

**Figure 4** **Comparison of mRNA expression levels of candidate genes in decidua tissues.** $n = 3$, mean $\pm$ SD, and independent sample T-test was used for comparison among groups.

system development, plasma membrane part, and sequence-specific DNA binding were the most related GO terms. In the pathway enrichment analysis, the DMRs were mainly involved with CAMs, type I diabetes mellitus, and ECM-receptor interaction. In addition, we proposed that aberrant methylation of novel genes, *ABR*, *ALCAM*, *HLA-E*, *HLA-G*, and *ISG15*, may be crucial for embryonic progress and development and that these may serve as candidate genes closely related to RM.

Among the top ranked genes in the MVPs, *ABR* is likely to be the candidate loci of RM. ABR, a regulator of Rho-family small GTPases, has been proven to have key roles during mitotic processes in human embryonic stem cells (hESCs) (*Ohgushi et al., 2017*). We speculate that abnormal methylation of ABR may result in fetal growth retardation by influencing the mitotic processes of hESCs leading to RM.

Based on the results of our analysis of 1,799 DMRs screened using the decidua samples from patients with RM and healthy pregnant women, *ISG15* was found to be the most relevant gene in RM. ISG15 is one of the several proteins induced by conceptus-derived type I or II interferons (IFNs) in the uterus and is implicated as an important factor in determining uterine receptivity to embryos in ruminants (*Zhao et al., 2016*). Further, *ISG15* is involved in early bovine embryonic development and regulates *IFNT* expression in the blastocyst (*Zhao et al., 2016*). Thus, it is a candidate gene playing an important role during pregnancy through fetal growth.

To more broadly explore the potential function of the genes by the RM-related DMRs, we conducted a GO enrichment analysis. The GO enrichment analysis revealed that

the DMRs were significantly enriched in response to system, multicellular organismal, and anatomical structure developments, the function of which is mostly performed by related genes, such as *HLA-G* and *HLA-E*. HLA-G belongs to the non-classical HLA class I antigens and is a tolerogenic molecule that acts on the cells related to both innate and adaptive immunities (*Verloes et al., 2017*). Besides its immunosuppressive function in transplantation, *HLA-G* expression is involved in implantation and protection of the semi-allogeneic fetus from the maternal immune system (*Carosella et al., 2008*; *Hunt et al., 2005*; *Rebmann, Wagner & Grossewilde, 2007*). Studies have proposed that *HLA-G* plays an important role in early embryonic development (*Yao et al., 2014*). HLA-E products (class Ib human leukocyte antigens) act in the immunology of human reproduction as modulators of the maternal immune system during pregnancy (*Gelmini et al., 2016*). Recent studies have shown the functions of the HLA-E molecule and possible interactions with HLA-G. The relevance of *HLA-G* and *HLA-E* expression in the maternal–fetal interface seems to be regarding the inhibition of NK cell-mediated lysis and possible influence on cytokine profiles (*Gelmini et al., 2016*). Pregnancy is a condition where women undergo major physiological and immunological alterations (*Mishra et al., 2019*), which are likely to be influenced or controlled by abnormal methylation levels of HLA-G and HLA-E.

A KEGG pathway enrichment analysis was employed to visualize the DMRs enriched for any pathways. CAMs, type I diabetes mellitus, and ECM-receptor interaction were found to be the frequently enriched pathways with several enriched genes, suggesting the potential roles of these genes in RM. For these pathways, *ALCAM*, *HLA-G*, and *HLA-E* were the most related genes. ALCAM is a member of the neuronal immunoglobulin-like domain superfamily of CAMs and promotes cell adhesion and signaling (*Corbel et al., 1996*; *DeBernardo & Chang, 1996*). It has been elucidated that *ALCAM* is required for proper nephrogenesis and functions downstream of *FZD3* during embryonic kidney development (*Cizelsky & Tata, 2014*). Thus, ALCAM is thought to play an important role in part of embryonic development and is also associated with fetal growth abnormalities.

These findings show that the abnormal methylation pattern of candidate genes may affect the stability of normal pregnancy and participate in the mechanisms that lead to RM. However, additional genetic and environmental factors might also play a role in the methylation patterns.

We detected the mRNA expression levels of the candidate genes, *ISG15*, *ABR*, *HLA-G*, *HLA-E*, and *ALCAM*. The association between the candidate genes and RM has been described earlier in this manuscript. The mRNA expression levels of the candidate genes were significantly higher in the RM group than in the control group. We speculate that the occurrence of RM might be related to the increase in mRNA expression levels of *ISG15*, *ABR*, *HLA-G*, *HLA-E*, and *ALCAM* in the decidua. In addition, the methylation level of *HLA-G* and *HLA-E* was lower in the RM group than in the control group. Hypermethylation inhibits gene expression. Therefore, we proposed that a decrease in the methylation level of HLA-G and HLA-E may increase mRNA expression, while an increase in the maternal *HLA-G* and *HLA-E* mRNA expression levels may affect fetal formation and development through an immune response and ultimately lead to RM.

## CONCLUSIONS

In conclusion, this study first analyzed the DNA methylation status and mRNA expression levels of *ISG15*, *ABR*, *HLA-G*, *HLA-E*, and *ALCAM* in decidua samples from patients with RM and healthy pregnant women. These five novel genes, all relevant to embryonic development, are likely to play a significant role in RM. Changes in the methylation and mRNA levels of these five genes may lead to the same abnormal embryonic gene expression, resulting in the blockage of embryonic or fetal formation and development and eventually leading to RM. Therefore, we hypothesized that *ISG15*, *ABR*, *HLA-G*, *HLA-E*, and *ALCAM* may be potential candidates for the progress and development of RM, which may also serve as new targets in the diagnosis of RM. While our results provide a direction for future research, limitations still exist in the present study. There is a possibility that this abnormal methylation is not the cause but a consequence of the defect that leads to RM. Further studies using large sample sizes are needed to validate the biological functions and molecular mechanism of these genes.

## ACKNOWLEDGEMENTS

We thank all the members for their generous participation.

### Funding

This work was supported by the National Key Research and Development Program of China (No. 2018YFC1002801) and the National Natural Science Foundation of China (No. 81771655 and No. 81571503). The funders had no role in study design, data collection and analysis, decision to publish, or preparation of the manuscript.

### Grant Disclosures

The following grant information was disclosed by the authors:
National Key Research and Development Program of China: 2018YFC1002801.
National Natural Science Foundation of China: 81771655, 81571503.

### Competing Interests

The authors declare there are no competing interests.

### Author Contributions

- Li Pi and Zhaofeng Zhang analyzed the data, conceived and designed the experiments, performed the experiments, prepared figures and/or tables, authored or reviewed drafts of the paper, and approved the final draft.
- Yan Gu, Xinyue Wang and Jianmei Wang conceived and designed the experiments, prepared figures and/or tables, authored or reviewed drafts of the paper, and approved the final draft.
- Jianhua Xu and Junwei Liu analyzed the data, conceived and designed the experiments, performed the experiments, prepared figures and/or tables, and approved the final draft.

- Xuan Zhang and Jing Du conceived and designed the experiments, performed the experiments, authored or reviewed drafts of the paper, and approved the final draft.

## Human Ethics

The following information was supplied relating to ethical approvals (i.e., approving body and any reference numbers):

The Shanghai Institute of Planned Parenthood Research Clinical Ethics Review Board approved this research (PJ2015).

## Microarray Data Deposition

The following information was supplied regarding the deposition of microarray data:

Microarray data is available at GEO: GSE141298 and figshare: Pi, Li (2019): A genome-wide screening of DNA methylation in decidua samples measured by HumanMethylation450 array. figshare. Dataset. https://doi.org/10.6084/m9.figshare. 8253128.v2.

## Data Availability

The raw measurements are available in the Supplemental File.

## Supplemental Information

Supplemental information for this article can be found online at http://dx.doi.org/10.7717/ peerj.8196#supplemental-information.

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
