# Peer review of "DNA methylation profiling in recurrent miscarriage"

_PeerJ, doi:10.7717/peerj.8196_

## Round 0.1 · original submission · Major Revisions

As you will see from the reviewers below, there are several key issues about the presentation of the data that must be addressed to be acceptable. Of particular note, the manuscript is more a summary of the technical details and finding and does not contextualize the work (revision of both intro and discussion).

You will note that reviewer two has some serious considerations about the power of the study, how the data was controlled, the inclusion criterion, etc. These are important issues that must be addressed in the revision to be acceptable for publication.

Reviewer 1 ·

Basic reporting

The manuscript titled “DNA methylation profiling in recurrent miscarriages” by Pi et al. seems to be technically sound with extensive experimental performance. The authors have presented an important research gap in light of recurrent miscarriages and how DNA methylation has potential in shedding light on the etiology of recurrent miscarriages. The authors have provided adequate background and literature referenced. However, certain issues needs to be resolved.

In line 152 and 153, the authors mentioned that those genes showed differential methylation levels. Is the differences in methylation level with respect to other genes or are they talking about the differential methylation levels of these genes with respect to the cases and control?

In line 205, the authors can rephrase the sentence as “…collected from 15 RM patients and 15 control.” The cases are the patients and the controls are health normal individuals.

In line 220, the authors can replace the word “relevancy” by “relevance”

In general, English language used needs to be improved.

Experimental design

In any good research paper, aims and objectives are clearly stated in the introduction section. After going through the manuscript, in the introduction section from line 64 till line 72, the authors seemed to have stated their aim. However, that section seems to be more of stating the methodologies and technologies used rather than stating the importance why this research is needed and how this research will be helpful. The authors need to justify their study and state clear aims and objectives.

Methylation is one of the dynamic mechanism which is likely to be vary with age, gestation, disease profile and medication. In materials and method section, the selection of cases should be more elaborated and justified. Also the given criteria for selecting the controls is not sufficient. Has the authors matched for age, gestation, parity, ethnicity?

In line 90 what does the authors mean by “…minced into small fragments and stored in RNAlater tissue collection solution”. Is this a standard protocol? if yes, the protocol needs to be explained more elaborately and give reason as to why this step is important.

The sample size seems to be small, but that cannot be changed now. So, the authors should at least be able to justify the sample size in light of published literature.

Validity of the findings

Discussion should be more subjective and merely not talking about the software used or supporting/constricting the results obtained. Differential methylation (if I understand correctly) between cases and controls with respect to the five genes is a good outcome. However, the so observed methylation could be cause or consequence of RM. So the authors need to discuss in light of this. The authors may go through a paper by Mishra et al., 2019 for the same (Differential global and MTHFR gene specific methylation patterns in preeclampsia and recurrent miscarriages: A case-control study from North India; Elsevier Gene).

Reviewer 2 ·

Basic reporting

The Results and the Discussion section should be more descriptive. The current version is a direct report of the outcomes of the bioinformatic tools without much detail.

The article misses any context of previous studies in this field. Is this the first study looking at RM methylation data? If not, provide the outcomes of previous studies in different population. Also, include whether the findings of the current study are in accordance or not with prior studies.

Experimental design

The study does not have enough power? The sample size is too small.

The inclusion and exclusion criteria are not explained.

Whether the two study groups were matched for factors (especially maternal age) other than recurrent miscarriage (RM) are not clearly indicated.

The article does not mention about Ethical approval.

Validity of the findings

The methylation data was generated from the arrays and then differential methylation analysis done. But no validations (technical or biological) are performed. It is important to validate the findings of the differential methylation analysis.

Additional comments

The present manuscript identifies biological markers and processes involved in recurrent miscarriage (RM) by differential methylation analysis of CpGs in 450K array. The study reports some significant findings but a lot of important details are missing from the manuscript.

1. Was the study approved by the ethics committee? If so, please provide the details.
2. Given the incidence of RM (1-5%), the sample size of the study appears to be too small for differential methylation analyses. Does the study have enough power? Please include the sample size calculations in the manuscript.
3. How was the differential methylation analysis validated?
4. Do mention the inclusion and exclusion criteria in detail.
5. Please provide the demographic details of the samples in a table.
6. Were the two groups matched for maternal age?
7. Did the control patients had any previous pregnancies or were they pregnant for the first time? Also, did all the enrolled pregnancies resulted in live births?
8. The Results and the Discussion section should be more descriptive. The current version is a direct report of the outcomes of the bioinformatic tools without much detail.
9. Where were the methylation variable positions (MVPs) located in the genes? Did they come from CpG islands or non CpG islands? Were the MVPs located in promoters for the genes showing differential expression?
10. Is this the first study looking at RM methylation data? If not, provide the outcomes of previous studies in different population. Also, include whether the findings of the current study are in accordance or not with prior studies.
11. English language needs to be improved throughout the manuscript.

---

## Round 0.2 · accepted · Accept

The revised manuscript is greatly improved and addresses the major concerns brought up by the reviewers initially.

The only remaining concern I have is the the quality of the images in figures 2, 3, and 4. The text is minute and pixelated. High resolution images need to be provided for final publication.

Reviewer 1 ·

Basic reporting

The authors have satisfactorily incorporated all the suggestions made from the previous review

Experimental design

The authors have satisfactorily incorporated all the suggestions made from the previous review

Validity of the findings

The authors have satisfactorily incorporated all the suggestions made from the previous review